# The Transcriptome and Metabolome Reveal the Potential Mechanism of Lodging Resistance in Intergeneric Hybrids between *Brassica napus* and *Capsella bursa-pastoris*

**DOI:** 10.3390/ijms23094481

**Published:** 2022-04-19

**Authors:** Libin Zhang, Liyun Miao, Jianjie He, Huaixin Li, Maoteng Li

**Affiliations:** 1College of Life Science and Technology, Huazhong University of Science and Technology, Wuhan 430074, China; libinzhang@hust.edu.cn (L.Z.); miaoliyun428@sxtcm.edu.cn (L.M.); jianjie_he@hust.edu.cn (J.H.); huaixin_lll@hust.edu.cn (H.L.); 2Hubei Bioinformatics & Molecular Imaging Key Laboratory, Department of Bioinformatics and Systems Biology, College of Life Science and Technology, Huazhong University of Science and Technology, Wuhan 430074, China; 3College of Basic Medical Sciences, Shanxi University of Traditional Chinese Medicine, Jinzhong 030619, China

**Keywords:** *Brassica napus*, lignocellulose, transcriptome, differentially expressed genes, metabolome

## Abstract

Lodging is one of the main reasons for the reduction in seed yield and is the limitation of mechanized harvesting in *B. napus*. The dissection of the regulatory mechanism of lodging resistance is an important goal in *B. napus*. In this study, the lodging resistant *B. napus* line, YG689, derived from the hybridization between *B. napus* cv. Zhongyou 821 (ZY821) and *Capsella bursa-pastoris*, was used to dissect the regulation mechanism of hard stem formation by integrating anatomical structure, transcriptome and metabolome analyses. It was shown that the lignocellulose content of YG689 is higher than that of ZY821, and some differentially expressed genes (DEGs) involved in the lignocellulose synthesis pathway were revealed by transcriptome analyses. Meanwhile, GC–TOF–MS and UPLC–QTOF–MS identified 40, 54, and 31 differential metabolites in the bolting stage, first flower stage, and the final flower stage. The differential accumulation of these metabolites might be associated with the lignocellulose biosynthesis in *B. napus*. Finally, some important genes that regulate the metabolic pathway of lignocellulose biosynthesis, such as *BnaA02g18920D*, *BnaA10g15590D*, *BnaC05g48040D,* and NewGene_216 were identified in *B. napus* through the combination of transcriptomics and metabolomics data. The present results explored the potential regulatory mechanism of lignocellulose biosynthesis, which provided a new clue for the breeding of *B. napus* with lodging resistance in the future.

## 1. Introduction

*B. napus* is one of the most important oil crops in the world, and the stem tissue is an advantageous energy material for the development of biodiesel. Crop lodging was mainly induced because of stem fall caused by strong winds and other factors, as well as root fall caused by poor root anchorage strength [1]. Stem strength, i.e., the bending and/or breaking strength of the culm, is important for stem lodging resistance [2]. The stem strength of crops is primarily determined by plant architecture (morphological traits and anatomical structure) [3]. In particular, the anatomical structure is a consequence of plant growth and development at the cellular level, such as cell division, cell growth, and cell spatial arrangement, and it is closely related to environmental factors. For example, the stem with a large diameter and a thick cell wall could increase the bending strength and lodging resistance of wheat, barley, and rice stems [2,4,5]. Plant height is considered to be an important morphological feature affecting crop lodging, but it is not the main factor determining crop lodging [6,7]. Lignocellulose is a major component of the mechanical strength of crop stems [8], and its content is related to the lodging resistance of crops [3,5,9]. Lignin and cellulose determine the mechanical strength of the stem [10], while the accumulation of starch could also increase the flexural strength and hardness of the stem [11]. Peng et al. [12] and Berry et al. [1] found that the increase in lignin and hemicellulose content could enhance the stalk strength and lodging resistance of wheat.

Metabolomics provides a powerful way to study the metabolic phenotype of *Brassica* species [13,14,15,16]. Tan et al. [13] identified the dynamic metabolic changes from both seeds and silique walls that occur during oil accumulation by using gas chromatography coupled with mass spectrometry (GC–MS) and demonstrated that the oil content was independent of leaf photosynthesis and phloem transport during oil accumulation, but it required the metabolic influx from the silique wall. Kortesniemi et al. [16] investigated NMR metabolomics of ripened and developing oilseed rape (*B. napus*) and turnip rape (*B. rapa*) and found differences in the major lipids and the minor metabolites between the two species. High-throughput sequencing is one of the important tools used to study important agronomic traits and gene expression regulation in *B. napus* [17,18,19,20,21]. For example, Li et al. [18] identified 71 candidate genes with stem lodging-related traits through the genome-wide association study (GWAS) method, and constructed a gene co-expression network based on transcriptome sequencing, which revealed the genetic basis of stem lodging traits in *B. napus*. Kawakatsu et al. [21] reported the phenotypic and transcriptomic landscape of 61 rice (*Oryza sativa*) accessions with highly diverse below-ground traits grown in an upland field, which provided a useful resource for understanding the genomic and transcriptomic bases of phenotypic variation under upland field conditions. The metabolomics analysis is the quantitative detection of all metabolites and their biochemical states in specific organisms or tissues. Moreover, high-throughput sequencing has been combined with metabolomics data to explore the complex regulatory metabolism networks [22,23,24,25,26,27,28,29]. For example, Guo et al. [29] characterized the acclimation of maize (*Zea mays* B73) to drought and cold stresses using physiological measurements and comparative transcriptomics combined with metabolomics during the stress treatments and recovery stages, which presented a model showing that the plant response to the combined stress is multi-faceted and revealed an ABA-dependent maize acclimation mechanism to the stress combination. However, there is a lack of a systematic investigation of the lignocellulose-related gene expression networks through the combination of different omics technologies.

Interspecific and intergeneric hybridization are widely used to create the new germplasm with valuable characteristics. Zhongyou 821 (ZY821) is a winter rapeseed variety with a high yield that is multi resistant with a wide adaptability, which was cultivated in China for many years [30]. Park [31] found that the content of erucic acid in the rapeseed oil of *C. bursa*-*pastoris* was significantly lower than that of other cruciferous plants. Moreover, *C. bursa*-*pastoris* displayed high resistance to black spot disease and *Sclerotinia sclerotiorum* [32,33] and could endure cold, salt, and drought stress [34], which was probably related to its lignified or wooden stems. Furthermore, the hybridizations between *B. napus* (2*n* = 38; female) and *C. bursa-pastoris* (male, 2*n* = 32) were performed in the fields by hand emasculation and pollination [33]. The stably inherited hybrids with the expected chromosome number and normal meiosis behaviors were screened. Coincidently, a hard stem material, YG689, was identified in its offspring population. YG689 was reported to contain high lignin content, a certain resistance to *Sclerotinia sclerotiorum,* and the obvious characteristics of its *C. bursa*-*pastoris* origin, such as wooden stems [33]. In addition, Shen et al. [35] obtained one doubled haploid (DH) population obtained from a cross between Y689 and *B. napus* var. Westar, and the QTL for the plant height (PH), branch initiation height (BIH), stem diameter (SD), and flowering time (FT) were obtained.

Although some studies had been carried out for the YG689, the molecular regulatory mechanism and metabolic pathway of lodging resistance still remains elusive in YG689. In the present study, the regulatory mechanism for the stem lodging resistance of YG689 was studied by combining the anatomical structure, comparative transcriptomic and metabolomic analyses. The present results shed light on the molecular regulatory pathways in lignocellulose biosynthesis, which provides a new clue for the breeding of *B. napus* with lodging resistance in the future.

## 2. Results

### 2.1. The Anatomical Structure and Lignocellulose Analysis of ZY821 and YG689

The stems of ZY821 and YG689 in the early flowering and maturation stages were used for the morphological and anatomical analyses. As shown in Figure 1C, the stem strength and dry weight of YG689 were significantly higher than that of ZY821. These results indicate that the YG689 had stronger stem lodging resistance than that of ZY821. To explore the difference in stem anatomical structure, the stems of YG689 and ZY821 were evenly divided into five sections from bottom to top (Figure 1B). It was revealed that the lateral distribution of lignin in YG689 was obviously wider than that of ZY821 (Figure 2). The stem anatomical structures between YG689 and ZY821 at the maturation stage were also compared, and similar results were obtained (Appendix A).

To further analyze the difference in the lignocellulose components in stems between YG689 and ZY821, the content of lignocellulose in the YG689 and ZY821 stems at the seedling, bolting and budding, early flowering, terminal flowering, and maturation stages were measured. It was revealed that the lignin content of YG689 was significantly higher than that of ZY821 at all stages (*p* < 0.05), and the cellulose content of YG689 was significantly higher than that of ZY821 at the bolting and budding stage and the terminal flowering stage (*p* < 0.01) (Figure 1D). The content of hemicellulose was also significantly higher in YG689 than that of ZY821 (*p* < 0.05) except at the seedling stage. The lignin monomers in YG689 and ZY821 stems at different developmental stages were further analyzed by using the GC–MS. It was found that the G monolignol was the main monomer, and the content in YG689 was higher than that of ZY821 at the seedling stage (Appendix A). Further analyses showed that the S monolignol also existed, but the content of the S monolignol was lower than that of the G monolignol in both YG689 and ZY821 (Appendix A).

### 2.2. Transcriptome Analysis of YG689 and ZY821

The transcriptome of YG689 and ZY821 for the stem at the initial flowering stage were analyzed by RNA-Seq technology, where 99,472,340 (SRA number: SRX1522099) and 134,015,130 (SRA number: SRX1142564) clean reads, after removing adaptor sequences and low-quality reads, were generated, respectively. It was revealed that 88.35% (87,884,725) of YG689 and 90.60% (121,415,423) of ZY821 clean reads were mapped to the *B. napus* genome, among which 79.08% (78,663,759) and 61.44% (82,335,196) of clean reads were uniquely mapped to the *B. napus* genome. The trinity program was then applied for the de novo assembly of the clean reads, and 2697 novel genes were identified by comparing the assembled transcripts with the genome annotation information of *B. napus*. The genes were then annotated to the COG, NR, Swiss-Prot, GO, and KEGG databases (Appendix A). Among these novel genes, 339 were annotated by COG (12.57%), 1509 were annotated by GO (55.95%), 418 were annotated by KEGG (15.50%), 1348 (49.98%) were annotated with Swiss-Prot, and 2284 (84.69%) were annotated with NR.

To reveal the global differential gene expression profiles between YG689 and ZY821, the DEGs, by setting the gene expression level of ZY821 as a control, were analyzed, and the up- or down-regulated genes in YG689 were observed. In total, 8644 DEGs (3818 up-regulated and 4826 down-regulated genes) were identified (Figure 3A). Thirteen DEGs were randomly selected for RT-qPCR validation. The majority of the expression trends of selected DEGs were consistent with the RNA-seq results (Figure 3B), which indicated the reliability of transcriptome sequencing results. The DEGs between YG689 and ZY821 were further annotated with Gene Ontology terms, which could be classified into three categories and 51 subcategories (Appendix A). For example, the “cellular process”, “metabolic process”, and “single-organism process” in the biological process were enriched in most of the DEGs. The DEGs were also analyzed by the Cytoscape Enrichment Map based on the GO annotation, and a total of 4873 DEGs were enriched in the biological process category (Appendix A). The overlapping terms between up-regulated and down-regulated genes were “metabolic process”, “response to stimulus”, “biological regulation”, “developmental process”, “localization” and “cellular process”. Importantly, we found that the up-regulated DEGs in YG689 were mainly assigned to the “rhythmic process”, “cellular component biogenesis”, and “signaling” terms, which suggested the activities of the cell metabolism and signal transductions were much more active in YG689 than in ZY821. To better understand the biological functions of DEGs between YG689 and ZY821, the DEGs were further assigned for the KEGG analysis (Appendix A with the listed top-20 pathways) and some important pathways associated with lignocellulose synthesis were found, including “arginine and proline metabolism”, “cysteine and methionine metabolism”, “carotenoid biosynthesis”, and “glutathione metabolism”. Moreover, the DEGs involved in these KEGG pathways were localized on the *B. napus* genome (Figure 4). Among them, 30 DEGs were found to be significantly associated with lignocellulose synthesis (Appendix A). For example, GH9B17, TPS1, UGE4, and GAUT13 were up-regulated in the “starch and sucrose metabolism” pathway in YG689. (Figure 5). Twelve DEGs related to the lignocellulose synthesis were randomly selected for further expression analyses in the stem of YG689, ZY821, and TN070 (another *B. napus* line with a soft stem) in bolting, early flowering, and final flowering stages. It was shown that nine DEGs were significantly up-regulated in the early-flowering stage of YG689, and the up-regulation of *BnaA09g42650* persisted to the final flowering stage and reached the highest value (Figure 3C). For example, *BnaA09g42650* is homologous to PRX17 in *Arabidopsis*, and the transcription factor AGL15 participates in the transition of vegetative growth to reproductive growth, as well as the formation of lignification tissues, by directly regulating PRX17. Therefore, we speculated that the hybridization between *C. bursa-pastoris* and ZY821 leads to the differential expression of lignin synthesis-related genes, which induced the stem traits of YG689.

Further analyses revealed that 624 DEGs could not be compared to the *B. napus* reference genome, 108 of which could be compared to the *C. bursa*-*pastoris* genome. Twenty-three DEGs have more than 40% of the homology rate and six DEGs have more than 80% of the homology rate. The 23 DEGs were subjected to KEGG analysis and five new genes (NewGene_6090, NewGene_3494, NewGene_1078, NewGene_2975, and NewGene_3483) were annotated. NewGene_6090 is homologous to Carubv10019973m (Vacuolar ATP synthase subunit A, VHA-A) of *C. bursa*-*pastoris* and participated in the oxidative phosphorylation pathway; NewGene_3494 is homologous to Carubv10003976m (RNA polymerase II large subunit, NRPB1) of *C. bursa*-*pastoris* and participates in the purine or pyrimidine metabolism pathway; NewGene_1078 is homologous to Carubv10011493m (2-oxoglutarate, 2OG) of *C. bursa*-*pastoris* and is involved in cysteine and methionine metabolism; NewGene_2975 is homologous to Carubv10001560m (syntaxin of plants 132, SYP132) of *C. bursa*-*pastoris* and is involved in SNARE interactions in the vesicular transport pathway; and NewGene_3483 is homologous to Carubv10008883m (beta glucosidase 40, BGLU40) of *C. bursa*-*pastoris* and is involved in starch and sucrose metabolism or phenylpropane biosynthesis. Notably, another 516 DEGs could not be compared to the *B. napus* genome and the *C. bursa*-*pastoris* genome, which indicates that these DEGs might be novel genes that derived from the chromosome exchange between *B. napus* and *C. bursa*-*pastoris*.

### 2.3. Metabolome Analysis of Stem of YG689 and ZY821

To reveal the differential metabolites in the stem of YG689 and ZY821, the metabolomics of the stems of YG689 and ZY821 in the bolting, early flowering, and terminal flowering stages were also analyzed by using the GC–TOF–MS and UPLC–QTOF–MS techniques. A total of 449 peaks were obtained by the GC–TOF–MS analysis, and the PCA score map of the metabolic spectrum showed that all the YG689 and ZY821 samples were in the 95% confidence interval (Appendix A). Twenty differential metabolites between YG689 and ZY821 at the bolting stage were identified (Appendix A), and the content of 19 metabolites in YG689 were lower than that of ZY821, which included saccharides (sucrose, L-threose, ribulose-5-phosphate, D- galactose, raffinose, and tagatose), amino acids and derivatives (L-valine, β-alanine and threo-beta-hyrdoxyaspartate) and other products of plant metabolism in *B. napus*. Twenty-six differential metabolites were identified in the early flowering stage (Appendix A), and the content of eight metabolites in YG689 is higher than that of ZY821, which included maleamate, elaidic acid, succinic acid, malonic acid, L-valine, 5, 6-dihydrouracil, erythrose and mannitol. Seven differential metabolites between YG689 and ZY821 at the final flowering stage were obtained (Appendix A), and four metabolites in YG689 were higher than of ZY821, including glutaconic acid, inosine, nornicotine, and leucrose.

The UPLC–QTOF–MS was also used to detect the differential metabolites between YG689 and ZY821, and 25, 36, and 39 differential metabolites were observed in the bolting, early flowering, and final flowering stages, respectively (Appendix A). In the bolting stage, the content of proline was increased in YG689, while 5-L-glutamyl-L-alanine, deoxyadenosine, and kaempferol were decreased in YG689 (Appendix A). In the early flowering stage, the content of L-threonate, aconitate, and (S)-2-hydroxyglutarate increased in YG689, while cytidine, guanosine, and cytidine 2′, 3′-cyclic phosphate decreased in YG689 (Appendix A). In the final flowering stage, the content of beta-D-fructose 6-phosphate, D-gluconate and L-isoleucine increased in YG689, while the content of N,N′-Diacetylchitobiose, alpha-linolenic acid, L-leucine and succinate decreased in YG689 (Appendix A).

Furthermore, the metabolites identified by GC–TOF–MS and UPLC–QTOF–MS were integrated together, and 40, 54, and 31 differential metabolites between YG689 and ZY821 in the bolting, initial flowering, and final flowering stages were identified, respectively. These differential metabolites were involved in nine important metabolic pathways (Appendix A), including glycolysis; the TCA cycle; the pentose phosphate pathway; fructose and mannose metabolism; galactose metabolism; phenylalanine, tyrosine, and tryptophan synthesis; starch and sucrose metabolism; glycine, serine, and threonine metabolism; and cysteine and methionine metabolism. For example, cis-aconitate and succinate were common differential metabolites in three developmental stages and were involved in TCA cycle pathways. Finally, the related metabolic pathway of lignocellulose synthesis was constructed by integrating DEGs and differential metabolites in the stem of YG689 and ZY821 (Figure 6), which consisted of 14 distinct metabolites and four important DEGs that might regulate the lignocellulose synthesis.

### 2.4. The Validation of DEGs Regulating the Metabolic Pathway of Lignocellulose Synthesis

The level of secondary metabolites is associated with gene expression, protein modification, and the response to environmental changes in the growth and development process of plant. A total of four DEGs (*BnaA02g18920D*, *BnaA10g15590D*, NewGene_216, and *BnaC05g48040D*) were identified to be the key enzyme genes and were involved in the regulation of lignocellulose synthesis (Figure 6). The RT-qPCR validation results were shown in Appendix A. For example, the expression of *BnaA02g18920D* (TPS1) increased, which may facilitate the production of Trehalose 6-phosphate (T6P) and the development of a hard stem in YG689. In addition, *BnaA10g15590D* is homologous with the ASP2 gene of *Arabidopsis* and participates in phenylalanine biosynthesis. *BnaA10g15590D* was up-regulated in YG689, which may stimulate the level of phenylalanine biosynthesis in the metabolic pathway of lignocellulose. Nevertheless, *BnaC05g48040D* (MS2) was down-regulated in YG689, which indicates the decrease in the phenylalanine and L-methionine levels in the metabolic pathway of lignocellulose. Meanwhile, some new genes that were expressed in YG689 derived from the *C. bursa*-*pastoris* genome were also identified. For example, NewGene_216 was predicted to be trehalose-phosphate synthase 7, which catalyzes UDP glucose to form trehalose-6-phosphate and participates in the synthesis of trehalose. Interestingly, although the expression of NewGene_216 increased, the content of trehalose decreased in YG689. We guessed that the content of trehalose was transformed by other pathways. Therefore, we speculated that these DEGs might affect the lignocellulose synthesis in the stem tissues of YG689, thus affecting the stem hardness in *B. napus*.

## 3. Discussion

Distant hybridization is an important way to create new germplasm resources [36]. For example, Liu et al. [37] transferred the clubroot-resistance genes from clubroot-resistant Chinese cabbages to *B. napus* by distant hybridization combined with embryo rescue. Gong et al. [38] obtained the hybrids with powdery mildew (PM) resistance through hybridization between the *B. napus* cultivar ‘Zhongshuang11′ and the PM-resistant *B. carinata.* Lodging is one of the main reasons that influences mechanized harvesting, which results in yield reduction [39]. Long et al. [40] reported that *Oryza longistaminata* has a strong stem and a high biomass productivity, and 12, 11, and 3 QTLs for the stem diameter (SD), stem length (SL), and breaking strength (BS), respectively, were obtained in the mapping population that was obtained between line 93–11 and *O. longistaminata*. In the previous studies, a germplasm of YG689 with high fertility and lodging resistance was obtained in the offspring that were derived from *B. napus* and *C. bursa-pastoris*, and further analyses revealed that YG689 had a high content of lignin [33]. Intensive research on the lodging resistance mechanism of YG689 is helpful to cultivate the new *B. napus* lines with lodging resistance characteristics.

Transcriptomic analyses have been widely used to find the candidate genes or study the regulatory mechanisms of important agronomic traits. For example, four candidate genes that regulate the lignin content were identified by the integration of GWAS and transcriptome sequencing, which provides insight into the genetic control of lodging and lignin in *B. napus* [17]. The unique QTLs for stem lodging-related traits (plant height, branch initiation height, and stem diameter) were found by Shen et al. [35] in *B. napus*, and some genes (including ESK1 and CESA6) involved in lodging resistance have been identified [18]. Li et al. [41] reported that the CESA9 conserved-site mutation could affect its association with the CESA complexes and cause the low-DP (degree of polymerization) cellulose synthesis, which significantly enhanced plant lodging resistance and biomass enzymatic saccharification in rice. In the present study, some DEGs that were closely related to lignocellulose synthesis were identified between ZY821 and YG689, and most of these DEGs were located in the main QTLs regions reported by Shen et al. [35]. Meanwhile, 624 DEGs could not be mapped to the *B. napus* genome, and 108 DEGs could be mapped to the *C. bursa*-*pastoris* genome. Some genes that were associated with the lignocellulose biosynthesis were included. For example, *BnaA09g42650* is homologous with PRX17 from *Arabidopsis*. PRX17 encodes a cell wall-localized class III peroxidase and is involved in lignified tissue formation. The transcription factor AGL15 (agamous-like15) directly regulates PRX 17 and participates in the transition from vegetative growth to reproductive growth and the formation of lignified tissue in *A. thaliana* [42]. Interestingly, 516 of 624 DEGs could be neither mapped to the *B. napus* nor mapped to the *C. bursa-pastoris* genome, which indicates that these new genes might derive from the genome recombination between *B. napus* and *C. bursa-pastoris.* Coincidently, Zhang et al. [43] identified 37 HE (homologous exchange) events in the progeny of a nascent allotetraploid (AADD) from two diploid progenitors of hexaploid bread wheat. The obtained HEs are highly enriched within gene bodies, giving rise to novel recombinant genes. Furthermore, the generation of chimeric genes was detected in the HEs of the allopolyploid *Brassica*, rice, *Arabidopsis suecica*, banana, and peanut [43], which provides a mechanism for the generation of new genes and new proteins in nascent allopolyploids.

Recently, the integration analysis of the transcriptome and metabolome has been applied to reveal the regulatory pathways of specific agronomic traits in *B. napus* [44,45,46,47,48,49]. For example, Jia et al. [44] performed the metabolomic and transcriptomic analyses of the yellow-flowered rapeseed cultivar ZS11 and the white-flowered inbred line WP, and it was shown that the white petal color in WP flowers is primarily due to decreased lutein and zeaxanthin contents, and BnNCED4b might play a key role in white petal formation. Tan et al. [45] investigated the gene expression profiles and metabolite content by the integration analysis of the transcriptome and metabolome in the seeds of *B. napus*. It was revealed that the expression of major carbohydrate metabolism-regulating genes was significantly correlated with carbohydrate content during seed maturation. In the present study, we integrated the DEGs and differential metabolites to construct a specific pathway of the lignocellulose metabolism in YG689. Notably, a total of 14 distinct metabolites and four DEGs were found to be involved in the regulation of lignocellulose synthesis. For example, the content of cis-aconitate and succinate was higher in the stem of YG689 than in ZY821 in three developmental stages. Importantly, cis-aconitate and succinate are involved in the TCA cycle pathway. Therefore, we speculate that the more active tricarboxylic acid cycle pathway promotes lignin synthesis in YG689 in comparison with ZY821. TPS1 is a gene coding for an enzyme that catalyzes the production of Trehalose 6-phosphate (T6P). It was reported that TPS1 is mainly expressed in axillary buds and the subtending vasculature, as well as in the leaf and stem vasculature [50]. A recent study showed that TPS1 is associated with the traits of plant height, peduncle length, and biomass in wheat [51]. Moreover, it was shown that the loss of TPS1 in *A. thaliana* impaired high-temperature-mediated hypocotyl growth [52]. Our study found that TPS1 was highly expressed in the stem tissue and up-regulated in YG689 compared with ZY821. Further integration analyses of the transcriptome and metabolome revealed that TPS1 was involved in the regulation of lignocellulose synthesis, which broadens our understanding of key genes regulating important agronomic traits of crops. In addition, as the most abundant pectic glycan, Homogalacturonan (HG) functions as a cell wall structural and signaling molecule essential for plant growth, development, and responses to pathogens [53]. GAUT13 was reported to de novo synthesize HG in the absence of exogenous HG acceptors [53]. In this study, GAUT13 was found to be up-regulated in YG689 (Appendix A), which suggests the functional significance of GAUT13 in the lignocellulose synthesis of *B. napus*. However, the exact mechanism of GAUT13 regulating the lignocellulose synthesis in *B. napus* needs to be further explored in the future. Lignocellulose is a major component of the mechanical strength of the crop stem tissue and is related to the lodging resistance of crops. The present results are not only useful for understanding the potential regulatory mechanism of lignocellulose biosynthesis, but they also suggest a novel strategy for breeding new varieties with lodging resistance traits, ultimately increasing rapeseed yield in the future.

## 4. Materials and Methods

### 4.1. Plant Materials

The seeds of YG689, ZY821, *C. bursa*-*pastoris,* and TN070 (the *B. napus* line with a soft stem) are provided by Professor Zaiyun Li of Huazhong Agricultural University, China. YG689 was selected from the successive cytological and fertility selection in the offspring derived from the hybridization between *B. napus* var. ZY821 and *C. bursa*-*pastoris* (Figure 1A). The materials were planted in the experimental field of Huazhong Agricultural University from 2013 to 2016. The seeds were generally sown in late September of the year and the plants and seeds were collected in early May of following year. The row spacing was 40 cm and the plant spacing was 20 cm.

### 4.2. Anatomical Structure and Lignocellulose Content Analysis of YG689 and ZY821

Stem samples of YG689 and ZY821 at the early flowering and mature stages with three replicates were evenly divided into five segments from the base to the top (Figure 1B). The middle part of the segment was taken for anatomic observation. Some of the transected materials were stained with 1% resorcinol (dissolved in 95% alcohol) for 2 min, then stained with concentrated hydrochloric acid for 1 min, and finally were observed and photographed by the stereomicroscope (Olympus MVX10, Japan). Meanwhile, the stem samples of YG689 and ZY821 at the seedling, bolting and budding, early flowering, final flowering, and maturation stages with three replicates were collected for the lignocellulose total content analysis. The seedling, bolting and budding, early flowering, final flowering, and maturation stages were 105, 142, 160, 178, and 215 days after sowing, respectively. The different developmental stages of stems were firstly blanched at 105℃ and dried at 60 ℃, and 5 mg of YG689 and ZY821 stalk powder were put into 10 mL test tubes for lignocellulose total content measurements, respectively. The acetyl bromide method was used to extract and detect lignin [54], and the content was calculated according to the Bouguer–Lambert–Beer law method [55]. The lignin monomer was prepared as previously described [56], and its contents were determined by GC–MS analysis. The cellulose and hemicellulose were extracted and detected according to the previous literature [57].

Fifteen whole plant materials of YG689 and ZY821 were randomly selected, and the stem length was measured after removing the roots. Furthermore, the 20-cm stems were cut from the stem base and were put on the stem strength tester (YYD-1) to measure the stem strength index. Stem fiber components were measured by near-infrared reflectance spectroscopy (NIRS) using NIR (FOSS, NIRS 5000) with WinISI software, according to previous reports [17]. Five random plants for YG689 and ZY821 were selected and their stems at 20 cm above the cotyledon scar were intercepted, dried, and ground into powder for measuring the fiber components. The phenotype values for acid detergent lignin (ADL), acid detergent fiber (ADF), and neutral detergent fiber (NDF) were speculated from NIRS spectra using NIRS calibrations for these traits, as described by Wei et al. [17].

### 4.3. RNA Extraction and Transcriptome Sequencing

Total RNAs were extracted from the stems of YG689 and ZY821 by the TriZol method (Invitrogen, Carlsbad, CA, USA), and the mRNAs were isolated from total RNA using Dynabeads oligo (dT) (Invitrogen). First- and second-strand cDNA were synthesized using Superscript II reverse transcriptase and random hexamer primers. Double-stranded cDNA was fragmented by nebulization and used to generate RNA-seq libraries, as previously described [58]. Three biological replicates of the cDNA libraries were sequenced using the Illumina Hiseq 2000 platform. The mRNA expression levels in YG689 and ZY821 were verified by using RT-qPCR. One microgram of total RNA was reverse-transcribed using SuperScript III reverse transcriptase and oligo (dT)18, according to the manufacturer’s instructions. The RT-qPCR reaction system was performed by using the TOYOBO SYBR^®^ R Green Realtime PCR Master Mix (code No. QPK-201) kit. RNA-seq data showed that the mRNA level of actin in ZY821 and YG689, or in each growth stage (the bolting stage, early flowering stage, and terminal flowering stage) was stable. Therefore, actin was set as the reference gene in RT-qPCR experiments. The primers for mRNA RT-qPCR are listed in Appendix A. The relative expression levels of these genes were measured by the 2^-ΔΔCt^ method using RT-qPCR.

### 4.4. Differentially Genes Expression and Function Enrichment Analysis

The differentially expressed genes (DEGs) between YG689 and ZY821 were identified using the expression levels of ZY821 transcripts as the control and were tested with the software package DESeq (version 1.12.3) [59] with a false discovery rate (FDR) of < 0.01 and a normalized fold change of ≥ 2. The GO enrichment analysis applied a hypergeometric test to find significantly enriched GO terms in DEGs comparing the genome background [60], where the calculating formula was the same as previously described [61], and the GO terms, with an adjusted *p*-value of 0.05, were defined as significantly enriched GO terms in DEGs. The enriched GO categories were visualized using the Cytoscape plug-in Enrichment Map (http://www.cytoscape.org/ (19/08/2021)). DEGs which could not be mapped to the *B. napus* reference genome [62] were aligned to the *C. bursa-pastoris* reference genome [63] to predict the origin and function of them.

### 4.5. Metabolome Analysis of YG689 and ZY821

Six biological replicates of each stem sample (0.05 g per sample) of YG689 and ZY821 in the bolting, early flowering, and terminal flowering stages were collected. A total of 36 stem samples were extracted for the GC–TOF–MS analysis, as previously described [64]. An Agilent 7890 GC system equipped with a Pegasus 4D TOFMS (LECO, St. Joseph, MI, USA) was used for the GC–TOF–MS analysis. Metabolite quantification was performed using a multiple reaction monitoring (MRM) method, as described [65]. Statistical significance was defined at *p* < 0.05, with highly significant values at *p* < 0.01. The VIP (variable importance in the projection) value (threshold > 1) of the first principal component of OPLS-DA model and the *p* value of the *t*-test (threshold 0.05) are used to identify the differentially expressed metabolites.

## Figures and Tables

**Figure 1 ijms-23-04481-f001:**
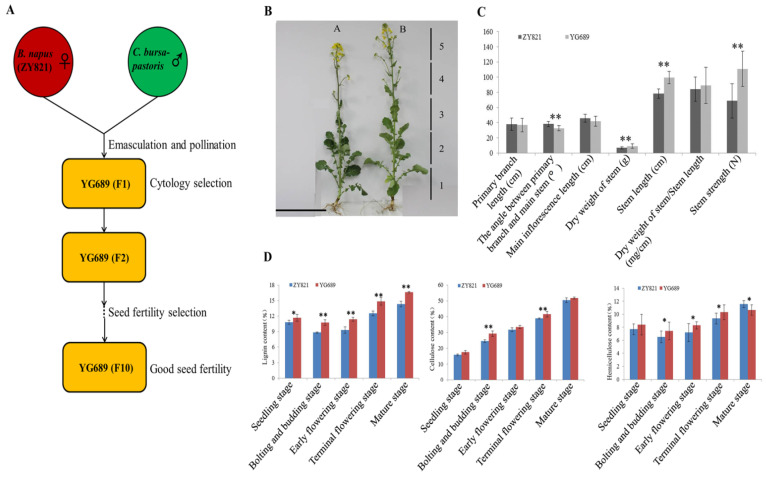
(**A**) The sketch map of YG689 generation from the hybridization of ZY821 and *C. bursa-pastoris*. (**B**) The morphology of ZY821 (Left) and YG689 (Right) at initial flowering stage. Bar = 50 cm. (**C**) The morphology analysis of ZY821 and YG689 at the mature stage (fifteen biological replicates). (**D**) Comparison of stem lignocellulose (lignin, cellulose, and hemicellulose) content of ZY821 and YG689 at five development stages (three biological replicates). * Significant at *p* < 0.05; ** significant at *p* < 0.01.

**Figure 2 ijms-23-04481-f002:**
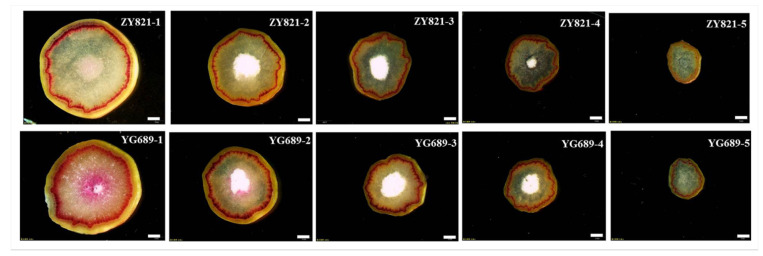
Transverse stem sections of ZY821 and YG689 at initial flowering stage. Stained red characteristics represent the distribution of lignin. The plant is divided into five parts from the base to the top, in turn, and labeled 1, 2, 3, 4, and 5. Bar = 2 mm.

**Figure 3 ijms-23-04481-f003:**
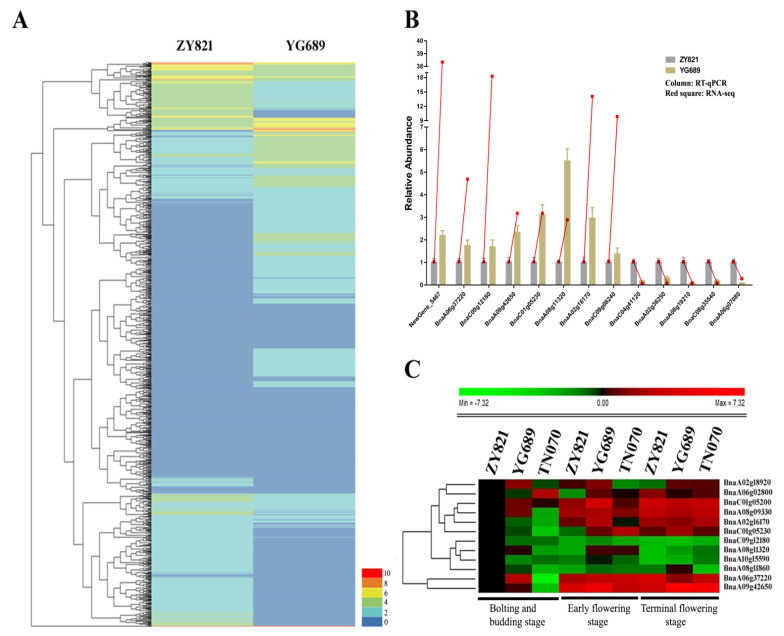
(**A**) Heatmap analysis of differentially expressed genes between YG689 and ZY821. RPKM (reads per kb per million reads) was used to calculate gene expression level. The color key (0, 2, 4, 6, 8, 10) represents FPKM normalized log(10) transformed counts. Red represents high expression and blue represents low expression. Each row represents a gene. (**B**) Validation of differentially expressed genes by RT-qPCR. Thirteen DEGs were randomly chosen for RT-qPCR validation using ZY821 transcript expression levels as the control. The relative expression levels of each gene are expressed as the fold change between ZY821 and YG689. The *B. napus* ACT 7 actin gene is used as an internal control. Histogram indicated the relative expression level between ZY821 and YG689 from RT-qPCR results and red square indicated the relative expression level between ZY821 and YG689 from RNA-seq. (**C**) Heatmap analysis of lignocellulose-related DEGs among ZY821, YG689, and TN070. The color key represents relative expression levels normalized log 2-transformed counts. Red represents high expression and green represents low expression. Each row represents a gene (*n* = 3).

**Figure 4 ijms-23-04481-f004:**
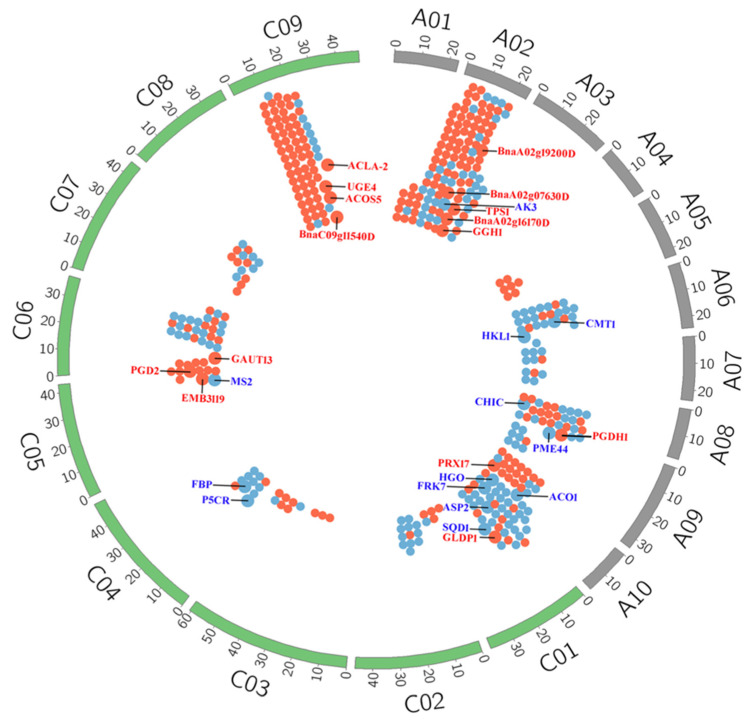
The distribution of the DEGs involved in KEGG pathways on *B. napus* genome. A01-A10 indicates 10 chromosomes of A genome in *B. napus*; C01-C09 indicates 9 chromosomes of C genome in *B. napus*. Red dots indicate up-regulated genes and blue dots indicate down-regulated genes. Large dots indicate 30 DEGs closely related to lignocellulose synthesis.

**Figure 5 ijms-23-04481-f005:**
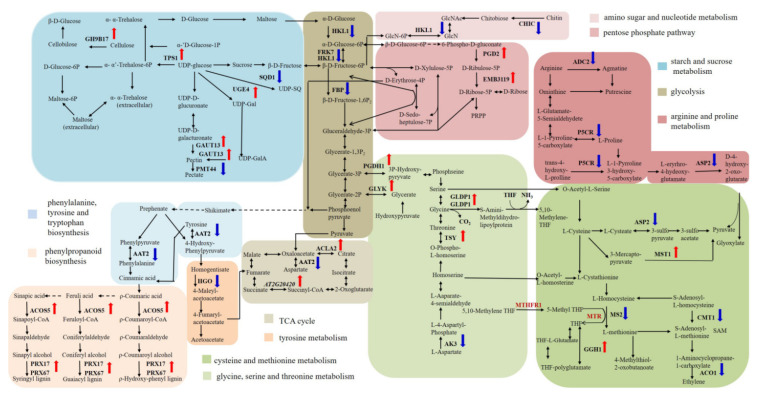
Analysis of metabolic pathways of the lignocellulose-related DEGs. The red arrows represent the up-regulated gene, and blue arrows represent the down-regulated gene.

**Figure 6 ijms-23-04481-f006:**
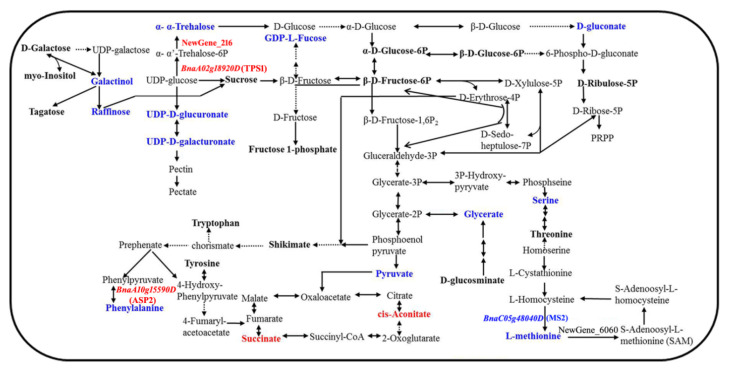
Metabolic pathway of lignocellulose by integrating DEGs, differentially accumulated metabolites between ZY821 and YG689. Red marks indicate up-regulated genes or up-accumulated metabolites in the pathway of lignocellulose synthesis. Blue marks indicate down-regulated genes or down-accumulated metabolites in the pathway of lignocellulose synthesis. NewGene_216 and NewGene_6060 are predicted new genes for *B. napus* by transcriptome analysis.

## Data Availability

Not applicable.

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
