# Peer review of "The Transcriptome and Metabolome Reveal the Potential Mechanism of Lodging Resistance in Intergeneric Hybrids between Brassica napus and Capsella bursa-pastoris"

_ijms, 2022, doi:10.3390/ijms23094481_

Round 1

Reviewer 1 Report

The manuscript investigates the molecular mechanisms causing resistance of an oilseed rape hybrid to lodging. The topic of the article is actual and it could be interesting for a wider spectrum of readers.  The text is readable, only small mistakes are presented. The used methods are sufficient. They are based on -omic analyses (RNA-seq, metabolome). The results were verified and evaluated by proper methods. The results are discussed with current literature.

Nevertheless, the manuscript has some weaknesses:

The number of analysed samples must be corrected to be clear. It is mentioned just for some methods, not for all. The used statistics is not mentioned. The time of sampling should be also better described (for example by illustrative photos, by days after sowing etc.). The current description of plant age is not sufficient.

Figures in the text have to be enlarged - especially Fig. 1C and 1D, 3B and 3C, 5, 6. They are too small and the text is not well readable.

Fig. 3B and S5 - These figures should compare the results from RNA-seq and RT-qPCR, but there are only data from qPCR (and RNA-seq data are presented differently). Please, compare the data properly.

The figures and the text could be easier if the genes are described by abbreviations than by accession numbers (especially in the schemes). The accession numbers from Arabidopsis are not transparent in Fig. 5 and they are confusing (why Arabidopsis when you measured oilseed rape). Similarly, l. 197-200 contain only Arabidopsis accession numbers, not Brassica.

Some references are not used in the text and they must be corrected (13-16, 46-49).

Please, correct the majority of present tense to past tense.

In the case of supplementary material, I suggest adding statistics to Table S1, because it is not possible to compare the results now. Moreover, the changes in the Table seem to be much more dynamic than that described in the text. The Dataset S1 should contain the first line with descriptions. Similarly, the supplementary figures need some legends to be clear (e.g. Fig. S4 is not clear). Table S6 should be in width.

The manuscript needs some small corrections:

l. 17 - "the lodging-resistant B. napus line YG689"

l. 101 - delete "the between"

Fig. 1 legend - it is not clear, how you measured stem strength. The legend should contain also a list of lignocellulose members in parentheses.

l. 130 - delete "highly"

l. 137 - correct the whole sentence

l. 157-158 - the percentage and number of genes are different than in the table.

Fig. 3 legend - Please, describe the numbers on the colour scale (1A), describe FPKM abbreviation. 1C is not clearly described (three repetitions of the same genotypes - probably different plant stages). You write that you compare 20 DEGs, but you show only 16.

l. 193 - phenylpropanoid biosynthesis is not listed in the supplementary dataset.

Fig. 4 legend - add the information about A and C and numbers in the scheme. It is not clear. Correct B. napus in italics.

l. 222-233 - Correct the missing gap in C. bursa-pastoris. Correct capital letter in Carubv... Write SYP132 in capitals.

l. 233 - Delete "and" after between

l. 237 - Correct "Metabolomic"

l. 254 - Glucosamine is not on the list - you have glucosaminic acid in Table S4. It was down-regulated as well as quinolinic acid and 2-methylglutaric acid.

 Fig. 6 legend - Delete "(A)." (not used). Correct B. napus in italics. Please, specify "different metabolites" differing from what?

l. 297 - Unfortunately, the connection with MS2 is missing in the scheme Fig. 6.

l. 301 - Please, correct the name of trehalose-phosphate synthase

l. 305 - Is it possible that some other genes from the same family can affect the synthesis? (gene redundancy)

l. 336 - I do not understand the end of the sentence.

l. 350 - Some references should be added.

l. 369 - Diurnal rhythm could affect the TCA cycle, too (it depends on the time of sampling).

RT-qPCR - Please, use the abbreviation according to MIQE guidelines. In the description of the method, add also the information that you checked the stability of the expression of the reference gene actin in both used variants and all growth stages (comparison e.g. by geNorm or by the stability of the expression from RNA-seq data). If the transcription is changing with genotype or with age, you cannot use it as the reference gene. Add the information how you calculated the results. Correct "Additional file 17" (l. 428).

l. 432 - Correct the reference.

Metabolomics - Please, specify the number and type of samples.

Author Response

Reviewer 1:

The manuscript investigates the molecular mechanisms causing resistance of an oilseed rape hybrid to lodging. The topic of the article is actual and it could be interesting for a wider spectrum of readers.  The text is readable, only small mistakes are presented. The used methods are sufficient. They are based on -omic analyses (RNA-seq, metabolome). The results were verified and evaluated by proper methods. The results are discussed with current literature.

Nevertheless, the manuscript has some weaknesses:

The number of analysed samples must be corrected to be clear. It is mentioned just for some methods, not for all. The used statistics is not mentioned. The time of sampling should be also better described (for example by illustrative photos, by days after sowing etc.). The current description of plant age is not sufficient.

Figures in the text have to be enlarged - especially Fig. 1C and 1D, 3B and 3C, 5, 6. They are too small and the text is not well readable.

Fig. 3B and S5 - These figures should compare the results from RNA-seq and RT-qPCR, but there are only data from qPCR (and RNA-seq data are presented differently). Please, compare the data properly.

The figures and the text could be easier if the genes are described by abbreviations than by accession numbers (especially in the schemes). The accession numbers from Arabidopsis are not transparent in Fig. 5 and they are confusing (why Arabidopsis when you measured oilseed rape). Similarly, l. 197-200 contain only Arabidopsis accession numbers, not Brassica.

Some references are not used in the text and they must be corrected (13-16, 46-49).

Please, correct the majority of present tense to past tense.

In the case of supplementary material, I suggest adding statistics to Table S1, because it is not possible to compare the results now. Moreover, the changes in the Table seem to be much more dynamic than that described in the text. The Dataset S1 should contain the first line with descriptions. Similarly, the supplementary figures need some legends to be clear (e.g. Fig. S4 is not clear). Table S6 should be in width.

The manuscript needs some small corrections:

  1. 17 - "the lodging-resistant B. napus line YG689"

  1. 101 - delete "the between"

Fig. 1 legend - it is not clear, how you measured stem strength. The legend should contain also a list of lignocellulose members in parentheses.

  1. 130 - delete "highly"

  1. 137 - correct the whole sentence

  1. 157-158 - the percentage and number of genes are different than in the table.

Fig. 3 legend - Please, describe the numbers on the colour scale (1A), describe FPKM abbreviation. 1C is not clearly described (three repetitions of the same genotypes - probably different plant stages). You write that you compare 20 DEGs, but you show only 16.

  1. 193 - phenylpropanoid biosynthesis is not listed in the supplementary dataset.

Fig. 4 legend - add the information about A and C and numbers in the scheme. It is not clear. Correct B. napus in italics.

  1. 222-233 - Correct the missing gap in C. bursa-pastoris. Correct capital letter in Carubv... Write SYP132 in capitals.

  1. 233 - Delete "and" after between

  1. 237 - Correct "Metabolomic"

  1. 254 - Glucosamine is not on the list - you have glucosaminic acid in Table S4. It was down-regulated as well as quinolinic acid and 2-methylglutaric acid.

 Fig. 6 legend - Delete "(A)." (not used). Correct B. napus in italics. Please, specify "different metabolites" differing from what?

  1. 297 - Unfortunately, the connection with MS2 is missing in the scheme Fig. 6.

  1. 301 - Please, correct the name of trehalose-phosphate synthase

  1. 305 - Is it possible that some other genes from the same family can affect the synthesis? (gene redundancy)

  1. 336 - I do not understand the end of the sentence.

  1. 350 - Some references should be added.

  1. 369 - Diurnal rhythm could affect the TCA cycle, too (it depends on the time of sampling).

RT-qPCR - Please, use the abbreviation according to MIQE guidelines. In the description of the method, add also the information that you checked the stability of the expression of the reference gene actin in both used variants and all growth stages (comparison e.g. by geNorm or by the stability of the expression from RNA-seq data). If the transcription is changing with genotype or with age, you cannot use it as the reference gene. Add the information how you calculated the results. Correct "Additional file 17" (l. 428).

  1. 432 - Correct the reference.

Metabolomics - Please, specify the number and type of samples.

Point-by-point response to the reviewer’s comments

Comments and Suggestions for Authors

The manuscript investigates the molecular mechanisms causing resistance of an oilseed rape hybrid to lodging. The topic of the article is actual and it could be interesting for a wider spectrum of readers.  The text is readable, only small mistakes are presented. The used methods are sufficient. They are based on -omic analyses (RNA-seq, metabolome). The results were verified and evaluated by proper methods. The results are discussed with current literature.

Nevertheless, the manuscript has some weaknesses:

The number of analysed samples must be corrected to be clear. It is mentioned just for some methods, not for all. The used statistics is not mentioned. The time of sampling should be also better described (for example by illustrative photos, by days after sowing etc.). The current description of plant age is not sufficient.

Response: Thanks for your suggestion. The information you mentioned has been added in the revised manuscript.

Figures in the text have to be enlarged - especially Fig. 1C and 1D, 3B and 3C, 5, 6. They are too small and the text is not well readable.

Response: Thanks for your suggestion. Fig 1 has been revised in the revised manuscript. Fig. 3B and 3C, Fig 5 and Fig 6 have been enlarged in the revised manuscript.

Fig. 3B and S5 - These figures should compare the results from RNA-seq and RT-qPCR, but there are only data from qPCR (and RNA-seq data are presented differently). Please, compare the data properly.

Response: Thanks for your suggestion. Fig. 3B and S5 have been revised in the revised manuscript.

The figures and the text could be easier if the genes are described by abbreviations than by accession numbers (especially in the schemes). The accession numbers from Arabidopsis are not transparent in Fig. 5 and they are confusing (why Arabidopsis when you measured oilseed rape). Similarly, l. 197-200 contain only Arabidopsis accession numbers, not Brassica.

Response: Thanks for your suggestion. We have revised Fig 5 in the revised manuscript. Moreover, Arabidopsis accession numbers have been deleted in the revised manuscript.

Some references are not used in the text and they must be corrected (13-16, 46-49).

Response: Sorry for the mistakes and they have been corrected in the revised manuscript.

Please, correct the majority of present tense to past tense.

Response: Thanks for your suggestion. Majority of present tense has been changed into past tense in the revised manuscript.

In the case of supplementary material, I suggest adding statistics to Table S1, because it is not possible to compare the results now. Moreover, the changes in the Table seem to be much more dynamic than that described in the text. The Dataset S1 should contain the first line with descriptions. Similarly, the supplementary figures need some legends to be clear (e.g. Fig. S4 is not clear). Table S6 should be in width.

Response: Thanks for your suggestion. Statistics has been added to Table S1 in the revised manuscript. Moreover, the supplementary dataset 1 has been revised in the revised manuscript. Finally, the figure legends of supplementary figures, including Fig S4, have been put in the section of “Supplementary Materials” in the revised manuscript. Table S6 has been revised in the revised manuscript.

The manuscript needs some small corrections:

  1. 17 - "the lodging-resistant B. napus line YG689"

Response: Thanks for your suggestion. We have corrected it in the revised manuscript.

  1. 101 - delete "the between"

Response: Thanks for your suggestion. We have corrected it in the revised manuscript.

Fig. 1 legend - it is not clear, how you measured stem strength. The legend should contain also a list of lignocellulose members in parentheses.

Response: Thanks for your suggestion. The measurement of stem strength has been added in the section of materials and methods in the revised manuscript. Moreover, a list of lignocellulose members in parentheses have been added in the legend of the revised manuscript.

  1. 130 - delete "highly"

Response: Thanks for your suggestion. We have corrected it in the revised manuscript.

  1. 137 - correct the whole sentence

Response: Thanks for your suggestion. We have corrected it in the revised manuscript.

  1. 157-158 - the percentage and number of genes are different than in the table.

Response: Sorry for the mistakes and they have been corrected in the revised manuscript.

Fig. 3 legend - Please, describe the numbers on the colour scale (1A), describe FPKM abbreviation. 1C is not clearly described (three repetitions of the same genotypes - probably different plant stages). You write that you compare 20 DEGs, but you show only 16.

Response: Sorry for the mistakes and they have been corrected in the revised manuscript.

  1. 193 - phenylpropanoid biosynthesis is not listed in the supplementary dataset.

Response: Sorry for the mistakes. “phenylpropanoid biosynethesis” has been deleted in the revised manuscript.

Fig. 4 legend - add the information about A and C and numbers in the scheme. It is not clear. Correct B. napus in italics.

Response: Thanks for your suggestion. We have corrected it in the revised manuscript.

  1. 222-233 - Correct the missing gap in C. bursa-pastoris. Correct capital letter in Carubv... Write SYP132 in capitals.

Response: Thanks for your suggestion. We have corrected them in the revised manuscript.

  1. 233 - Delete "and" after between

Response: Thanks for your suggestion. We have corrected it in the revised manuscript.

  1. 237 - Correct "Metabolomic"

Response: Thanks for your suggestion. We have corrected it in the revised manuscript.

  1. 254 - Glucosamine is not on the list - you have glucosaminic acid in Table S4. It was down-regulated as well as quinolinic acid and 2-methylglutaric acid.

Response: Sorry for the mistakes. We have corrected them in the revised manuscript.

Fig. 6 legend - Delete "(A)." (not used). Correct B. napus in italics. Please, specify "different metabolites" differing from what?

Response: Thanks for your suggestion. We have corrected them in the revised manuscript.

  1. 297 - Unfortunately, the connection with MS2 is missing in the scheme Fig. 6.

Response: Thanks for your suggestion. We have corrected it in the revised manuscript.

  1. 301 - Please, correct the name of trehalose-phosphate synthase

Response: Sorry for the mistake. We have corrected it in the revised manuscript.

  1. 305 - Is it possible that some other genes from the same family can affect the synthesis? (gene redundancy)

Response: Thanks for your suggestion. I think it is very possible. We will study the roles of homologous genes of BnaA02g18920D, BnaA10g15590D, BnaC05g48040D and NewGene_216 in the pathway of lignocellulose biosynthesis in the future.

  1. 336 - I do not understand the end of the sentence.

Response: Sorry for the mistake. We have deleted the end of the sentence.

  1. 350 - Some references should be added.

Response: Thanks for your suggestion. We have added the reference in the revised manuscript.

  1. 369 - Diurnal rhythm could affect the TCA cycle, too (it depends on the time of sampling).

Response: Thanks for your comment. We totally agree with your opinion.

RT-qPCR - Please, use the abbreviation according to MIQE guidelines. In the description of the method, add also the information that you checked the stability of the expression of the reference gene actin in both used variants and all growth stages (comparison e.g. by geNorm or by the stability of the expression from RNA-seq data). If the transcription is changing with genotype or with age, you cannot use it as the reference gene. Add the information how you calculated the results. Correct "Additional file 17" (l. 428).

Response: Thanks for your suggestion. We have used right abbreviation “RT-qPCR”. Moreover, the reference gene information and calculation information of RT-qPCR have been added in the revised manuscript (Materials and Methods section). "Additional file 17" has been deleted in the revised manuscript.

  1. 432 - Correct the reference.

Response: Sorry for the mistake. We have corrected it in the revised manuscript.

Metabolomics - Please, specify the number and type of samples

Response: Thanks for your suggestion. We have specified the number and sample type of metabolome in the “materials and methods” section of the revised manuscript.

Reviewer 2 Report

In this study, the molecular mechanism of lodging resistance in intergeneric hybrids between the Brassica napus and Capsella bursa-pastorisis is investigated. The experiment is well done and the authors present very interesting data. Therefore, I think that it is a suitable work to be published in IJMS

Minor point

Figure 4, What do A01, A02, C01, and the numbers below them mean? Please note them in the legend.

Figure 6, “(A)” is not required. The authors should indicate in the legend what the red and blue letters indicate. What does "different" mean? The authors should revise for clarity.

Line 301, Some text is missing in trehalose phosphate synthase 7.

Author Response

Reviewer 2:

In this study, the molecular mechanism of lodging resistance in intergeneric hybrids between the Brassica napus and Capsella bursa-pastorisis is investigated. The experiment is well done and the authors present very interesting data. Therefore, I think that it is a suitable work to be published in IJMS

Minor point

Figure 4, What do A01, A02, C01, and the numbers below them mean? Please note them in the legend.

Figure 6, “(A)” is not required. The authors should indicate in the legend what the red and blue letters indicate. What does "different" mean? The authors should revise for clarity.

Line 301, Some text is missing in trehalose phosphate synthase 7.

Point-by-point response to the reviewer’s comments

Comments and Suggestions for Authors

In this study, the molecular mechanism of lodging resistance in intergeneric hybrids between the Brassica napus and Capsella bursa-pastorisis is investigated. The experiment is well done and the authors present very interesting data. Therefore, I think that it is a suitable work to be published in IJMS

Minor point

Figure 4, What do A01, A02, C01, and the numbers below them mean? Please note them in the legend.

Response: Thanks for your question. A01-A10 indicated 10 chromosomes of A genome in B. napus; C01-C09 indicated 9 chromosomes of C genome in B. napus. The numbers below them mean chromosome length (Mb).

Figure 6, “(A)” is not required. The authors should indicate in the legend what the red and blue letters indicate. What does "different" mean? The authors should revise for clarity.

Response: Thanks for your suggestion. Red letters indicates up-regulated genes or up-accumulated metabolites. Blue letters indicates down-regulated genes or down-accumulated metabolites. "different" means differentially accumulated metabolites. We have revised them in the revised manuscript.

Line 301, Some text is missing in trehalose phosphate synthase 7.

Response: Sorry for the mistake. We have corrected it in the revised manuscript.

Reviewer 3 Report

The MS is devoted to the study of genetic characteristics  and diffremces of the  Brassica napus and Capsella bursa-pastoris plants to lodging resistance.
The authors have done an interesting work, which is of great interest for modern biology.
Some questions.
The quality of Figure 2B is such that it does not allow a full assessment of the conclusions described.
I suggest that the authors either improve its quality (the cut of the stem is very thick) or not use it at all in the article.

Despite the difficult work done by the authors, the conclusions are rather superficial.
It is necessary to discuss the results obtained more subtly in the discussion, all analyzes are reduced to an increase in the synthesis of lignocellulose.

Author Response

Reviewer 3:

The MS is devoted to the study of genetic characteristics  and diffremces of the  Brassica napus and Capsella bursa-pastoris plants to lodging resistance.

The authors have done an interesting work, which is of great interest for modern biology.

Some questions.

The quality of Figure 2B is such that it does not allow a full assessment of the conclusions described.

I suggest that the authors either improve its quality (the cut of the stem is very thick) or not use it at all in the article.

Despite the difficult work done by the authors, the conclusions are rather superficial.

It is necessary to discuss the results obtained more subtly in the discussion, all analyzes are reduced to an increase in the synthesis of lignocellulose.

Point-by-point response to the reviewer’s comments

Comments and Suggestions for Authors

The MS is devoted to the study of genetic characteristics  and diffremces of the  Brassica napus and Capsella bursa-pastoris plants to lodging resistance.
The authors have done an interesting work, which is of great interest for modern biology.
Some questions.
The quality of Figure 2B is such that it does not allow a full assessment of the conclusions described.
I suggest that the authors either improve its quality (the cut of the stem is very thick) or not use it at all in the article.

Response: Thanks for your suggestion. We have deleted Figure 2B in the revised manuscript.

Despite the difficult work done by the authors, the conclusions are rather superficial.
It is necessary to discuss the results obtained more subtly in the discussion, all analyzes are reduced to an increase in the synthesis of lignocellulose.

Response: Thanks for your suggestion. We have reorganized part of the discussion.
